# High Emissivity MoSi_2_-SiC-Al_2_O_3_ Coating on Rigid Insulation Tiles with Enhanced Thermal Protection Performance

**DOI:** 10.3390/ma17010220

**Published:** 2023-12-30

**Authors:** Xukun Yang, Yange Wan, Jiancun Li, Jiachen Liu, Mingchao Wang, Xin Tao

**Affiliations:** 1Key Lab of Advanced Ceramics and Machining Technology of Ministry of Education, School of Materials Science and Engineering, Tianjin University, Tianjin 300072, China; 2Beijing Institute of Astronautical Systems Engineering, Beijing 100076, China; 3School of Safety Science and Engineering, Civil Aviation University of China, Tianjin 300300, China; 4College of Science, Civil Aviation University of China, Tianjin 300300, China

**Keywords:** high emissivity coating, thermal protection system, alumina sol, flame heating experiment

## Abstract

High emissivity coatings with sol as the binder have the advantages of room temperature curing, good thermal shock resistance, and high emissivity; however, only silica sol has been used in the current systems. In this study, aluminum sol was used as the binder for the first time, and MoSi_2_ and SiC were used as emittance agents to prepare a high emissivity MoSi_2_-SiC-Al_2_O_3_ coating on mullite insulation tiles. The evolution of structure and composition at 1000–1400 °C, the spectral emissivity from 200 nm to 25 μm, and the insulation performance were studied. Compared with the coating with silica sol as a binder, the MoSi_2_-SiC-Al_2_O_3_ coating has better structural uniformity and greater surface roughness and can generate mullite whiskers at lower temperatures. The total emissivity is 0.922 and 0.897, respectively, at the wavelength range of 200–2500 nm and 2.5–25 μm, and the superior emissivity at a low wavelength (<10 μm) is related to a higher surface roughness and reduced feature absorption. The emissivity reduction related to the oxidation of emittance agents at a high temperature (−10.2%) is smaller than that of the silica-sol-bonded coating (−18.6%). The cold surface temperature of the coated substrate is 215 °C lower than the bare substrate, suggesting excellent thermal insulation performance of the coating.

## 1. Introduction

Advanced vehicles are subjected to severe aerodynamic heating environments during long-time flights and reentry. Thermal protection systems that are light weight, have no physical and chemical damage at high temperatures, and have reusability meet the requirements of advanced vehicles for flights, and they are non-ablative, long-term efficient, have high reliability, and have low maintenance costs [1,2,3]. The thermal protection system is mainly composed of ceramic insulation tiles and a high emissivity coating that can significantly enhance radiative heat transfer and reradiate heat away from the system during high-speed flights [4]. Therefore, it is important to increase the heat resistance and emissivity of the coating to further improve the thermal shielding and insulation performance of the system.

High emissivity coating is generally composed of a bonding phase, refractory fillers, and emissive agents. Compared with the traditional glass-bonded coating, silica-sol-bonded coating possesses better thermal shock resistance, higher emissivity, and room temperature curing properties and, therefore, has recently received extensive attention [5,6]. Super-fine silica colloidal materials are prepared through a sol–gel route, forming a three-dimensional network that surrounds the refractory particles and gives strength to drying during the green stage. When heating to higher temperatures, gel bonding is replaced by ceramic bonding through sintering, leading to much higher strength [7,8]. Guo et al. prepared a MoSi_2_-SiC-Al_2_O_3_ coating on a rigid ceramic tile with high bonding strength and contact damage resistance and thoroughly investigated the bonding mechanisms of silica sol at the temperature range of 600 °C to 1500 °C [9,10]. Xue et al. developed a silica-sol-bonded coating on aluminum silicate ceramic cloth, which presented a low solar absorption ratio and high emissivity [5]. At present, only silica sol has been applied as a binder phase in sol-bonded high-emissivity coatings. Though silica sol as a binder has the characteristics of high bonding strength and an easy preparation process, it also has certain drawbacks. On the one hand, silica sol will form a low melting point liquid phase at high temperatures, which greatly limits the application of the coating at temperatures higher than 1400 °C. On the other hand, the asymmetrical Si-O stretching vibrations of [SiO_4_] tetrahedra produce a significant absorption band centered at around 9 μm, resulting in a significant decrease in the coating emissivity at this band [11]. Therefore, aluminum sol is applied as the binder phase for the high emissivity coating in this study to improve the heat resistance of the coating and the emissivity of this band. The increase in the heat resistance of the coating can make it suitable for faster aircraft or key parts of the aircraft, such as the leading edge of the wing and the nose cone, and the increase in the emissivity can improve the heat protection ability of the insulation tile, thereby reducing the thickness of the insulation tile and the weight of the thermal protection system, which are crucial for the radiation heat protection system.

Aluminum sol has been applied as an inorganic binder in a wide variety of fields. It has been used as the binder in alumina-spinel refractory castables to replace cement and prevent the formation of phases with low melting points [12,13,14]. Although the bonding strength is relatively low, its thermal shock resistance is obviously better, and the strength retention rate is higher after thermal cycles for the absence of any thermal mismatching due to high compositional purity [4]. Moreover, aluminum sol is used as a catalyst adhesive in the petrochemical industry, automobile exhaust purification, and other applications. It can also be used as the forming binder for aluminum silicate fiber and ceramics [15,16]. Therefore, aluminum sol is a potential inorganic binder for the high emissivity coating in thermal protection systems, and the study of the effect of aluminum sol on the bonding ability, emissivity, and thermal protection performance of the high emissivity coating is of great significance.

In this study, aluminum sol is used as the bonding phase for high emissivity coating for the first time. The coating with silica sol as the binder (denoted as Si-based coating) and the coating with aluminum sol as the binder (denoted as Al-based coating) were prepared on rigid mullite insulation tiles, in which the composition of refractory fillers and emissive agents were the same. The structure and composition evolution of the two coatings were compared from room temperature to 1400 °C, and the interaction between aluminum sol and other components in the coating was revealed. The spectral emissivity of the two coatings was compared at the wavelength range of 200 nm–25 μm, and both the coatings after drying and the coatings after high-temperature heat treatment were studied. Finally, the thermal shielding and insulation ability of the aluminum-sol-bonded coating was verified via a flame heating experiment.

## 2. Materials and Methods

The substrates were commercially available mullite fibrous ceramics (Jiuwei Refractory Material Co., Ltd., Huzhou, China) with a density of 0.40 g/cm^3^ and compressive strength of 0.8 MPa. Specimens were polished to a 1200-grit finish on the surfaces and cleaned by a vacuum sweeper before coating deposition. The coating was fabricated by a slurry technique. Firstly, nano aluminum oxide (Al_2_O_3_, 30 nm, Far new Material, Suzhou, China), molybdenum disilicide (MoSi_2_, 3 μm, Eno Material, Liaocheng, China), silicon carbide (SiC, 0.5 μm, Eno Material, Liaocheng, China) and silica sol or alumina sol were mixed by ball milling for 1 h to form a homogeneous slurry. The ball-milling process used a planetary ball mill, alumina grinding ball with a diameter of 10 mm, and the ball material ratio is 2:1. Thereinto, MoSi_2_ and SiC were emissive agents, nano Al_2_O_3_ was the refractory filler, silica sol and alumina sol were the binder of the coating. The mass ratio of sol:Al_2_O_3_:MoSi_2_:SiC in the slurry was 20:6:2:3. The silica sol and alumina sol were bought from Jinghuo Technology Glass Co., Ltd., Dezhou, China, and their basic properties were listed in Table 1. Secondly, the slurry was deposited on the substrates by spraying, and the deposition amount of solid phase coating was 0.09 g/cm^2^. During the spraying process, air compressor was used to drive the pneumatic spray gun; the air pressure was 0.2 MPa, the nozzle diameter of the spray gun was 1 mm, and the spraying distance was 20 cm. Finally, the coated specimens were dried at 50 °C for 10 h in a drying oven. The coating with silica sol as binder was abbreviated as Si-based coating, and that using alumina sol was abbreviated as Al-based coating. The coatings were calcined at 1000 °C, 1200 °C, and 1400 °C for 1 h, and the calcined samples were tested to explore their composition, structure, and properties after high-temperature treatment.

Phase composition of the coatings was analyzed via X-ray Diffraction (XRD, D/Max-2500 Rigaku, Tokyo, Japan) with filtered Cu-Kα radiation. Scanning Electron Microscope (SEM, S-4800, Hitachi, Tokyo, Japan) was used for microstructural analysis. Thermogravimetric analysis was performed using a combined TGA/DSC instrument (STA-449C, Netzsch, Bavaria, Germany) with a heating speed of 10 °C/min in static air. The molecular structure of the coatings was identified by Fourier-transform infrared spectroscopy (FTIR, Nicolet iS5, Thermo Scientific, Waltham, MA, USA) in the wavenumber range of 4000 cm^−1^∼400 cm^−1^ in transmission mode via a KBr pellet method.

The integrating sphere reflectivity of the coatings was measured by a Fourier transform infrared spectrometer (Nicolet IS50, Thermo Fisher, Waltham, MA, USA) at the wavelength range of 2.5~25 μm and a UV-Vis-NIR Spectrophotometer (Lambda 750 s, Perkin-Elmer, Waltham, MA, USA) at the wavelength range of 0.2~2.5 μm. The spectral emissivity was obtained from the measured reflectivity based on the relation for materials, ε = *A* = 1 − *R* − *T*: where ε, *A*, *R*, and *T* being the emissivity, absorptivity, reflectivity, and transmissivity, respectively. For a perfect opaque body, the total emissivity can be derived from the reflectance spectrum according to Equations (1) and (2).
(1)$$\varepsilon_{T}=\frac{\int_{\lambda_{1}}^{\lambda_{2}}\left[1-R\left(\lambda \right)\right]P_{B}\left(\lambda \right)d\lambda }{\int_{\lambda_{1}}^{\lambda_{2}}P_{B}\left(\lambda \right)d\lambda }$$
(2)$$P_{B}\left(\lambda \right)=\frac{C_{1}}{\lambda^{5}\left(\exp\frac{C_{2}}{\lambda T}-1\right)}$$
where *λ* is the wavelength, *R*(*λ*) is the reflectance, *P_B_*(*λ*) is given by Planck’s law, *C*_1_ = 3.743 × 10^−16^ Wm^2^, *C*_2_ = 1.4387 × 10^−2^ mK.

The heat protection and insulation properties of the coatings were tested via a flame-heating experiment. The Al-based coating was prepared on two kinds of substrates with size of 100 mm × 130 mm × 8 mm and 100 mm × 130 mm × 15 mm, and the coating thickness is around 300 μm. The coating side of the sample was heated with a butane flame spray gun, the gas flow was adjusted to the maximum, and the nozzle of the spray gun was 100 mm away from the sample surface. The back temperature of the sample was recorded by a K-type thermocouple, which was attached to the backside of the sample and connected to a data acquisition device.

## 3. Results

### 3.1. Coating Structure

The coating structure of the Al-based coating and Si-based coating after drying are compared first. The surface and cross-sectional SEM images of the as-prepared Al-based coating are shown in Figure 1a,d and Figure 2a. The coating has a flat surface and a thickness of about 300 μm. The surface structure and thickness of the coating are uniform, and there is no obvious crack or hole. The high magnification view suggests that the particles in the coating stack together to form a porous coating structure. The surface and cross-sectional SEM images of the as-prepared Si-based coating are shown in Figure 2b and Figure 3a,d. The structure is uniform, and the coating thickness is about 300 μm, too. Compared with the Al-based coating, the Si-based coating shows a smoother surface and a denser structure. The high magnification view shows that the particles are surrounded by a continuous matrix and form a good combination. Silica sol is proven to be a better coating binder than alumina sol for the better binding capacity, which endows the coating with better cohesion strength and higher density [17,18]. The main reasons are as follows: first and foremost, the structural strength of the gel network of silica sol is higher than that of aluminum sol. Moreover, the silica sol possesses a lower particle size and higher solid content, which makes the cohesive phase form a more continuous and dense structure.

The surface SEM images of the Al-based coating after calcination at 1000–1400 °C are also shown in Figure 1. Some fine cracks appear on the surface when the coating is calcined at 1000 °C for 1 h (Figure 1b,e), which are about 1 μm wide. The cracks are not long or straight through the entire coating but meander and have crosslinked distribution on the surface. When the calcination temperature is 1200 °C (Figure 1c,f), the cracks on the coating surface are wider, and acicular crystals are observed in the high magnification view, which may be mullite whiskers, as speculated based on the coating composition. When the coating is calcined at 1400 °C (Figure 1g–i), the coating surface becomes uniform without cracks. The surface of the coating is completely covered by acicular crystals, and the surface area and roughness are significantly increased. The surface SEM images of the Si-based coating after calcination at 1000–1400 °C are shown in Figure 3. Numerous cracks appear on the coating surface when the coating is calcined at 1000 °C (Figure 3b,e); the crack distribution is uneven, and the width of the cracks is obviously larger than that in the Al-based coating after the same calcination treatment. The cracks are further deepened and widened when the calcination temperature increases to 1200 °C (Figure 3c,f), which may cause the coating to lose the mechanical properties required for safe service. When the coating is calcined at 1400 °C (Figure 3g–i), plenty of the glass bonding phase is generated in the coating, which makes the sharp crack edges disappear and the coating structure denser in the areas without cracks. The glass phases are formed by the melting of nano-silica in the silica sol, which is a new bonding phase with higher cohesive strength. Moreover, some needle-like crystals, which should be mullite whiskers, can be seen in the glass phase. The structural differences between the Al-based coating and Si-based coatings after high-temperature treatment are mainly in the following two aspects. On the one hand, the Al-based coating shows a more uniform surface structure, and the cracks are finer after high-temperature treatment compared to the Si-based coating. Since the binder phase in the Al-based coating possesses a lower binding effect and density, the moisture removal during calcination results in a more permeable structure, reducing the chances of crack generation and explosive spalling [13]. Some research revealed that the scale coating structure with a uniform crack network can relieve stress concentration at the interface between the coating and substrate, increase the strain tolerance of the coating, and improve the mechanical properties [19,20]. On the other hand, the temperature of the mullite whisker formation in the Al-based coating is lower, and the amount is larger, which gives the two coatings completely different structures after calcination at 1400 °C: the Al-based coating shows porous, high-roughness surfaces with crossed whiskers, while the Si-based coating shows a flat surface with a glass binder phase.

### 3.2. Coating Composition Evolution

The XRD patterns of the as-prepared Al-based coating and Si-based coating are shown in Figure 4a,b. Since the alumina gel and silica gel after drying are amorphous, the crystal phase composition of the two coatings is similar and mainly contains molybdenum disilicide (PDF#65-9392), silicon carbide (PDF#65-0360) and corundum (PDF#99-0036).

The XRD patterns of the Al-based coating and Si-based coating after calcined at 1000–1400 °C are also shown respectively in Figure 4a,b to clarify the composition and phase changes in the coatings at high temperatures. As for the Al-based coating calcined at 1000 °C, mullite is generated, and the diffraction peak intensity of MoSi_2_ decreases because part of MoSi_2_ has been oxidized (Reaction 1), and the oxidation product MoO_3_ has volatilized [21]. The mullite is the reaction product of nano Al_2_O_3_ (the refractory filler) and SiO_2_ (the product of Reaction 1) according to Reaction 2. When the coating is calcined at 1200 °C, the amount of mullite increases, the oxidation of SiC begins (Reaction 3), and cristobalite is detected. Research shows that the oxidation rate of SiC is strongly dependent on the reaction temperature and SiC particle size. Increasing the temperature to 1200 °C or higher causes the oxidation rate to increase and changes the amorphous SiO_2_ to the cristobalite phase [22]. When the calcination temperature is 1400 °C, the amount of mullite and cristobalite increases further, and the emissive SiC is completely oxidized. Therefore, the remaining MoSi_2_ is used as the only emissive agent at 1400 °C. The formation temperature of mullite determined by XRD (1000 °C) is lower than that of SEM (1200 °C), and the XRD results also prove that the acicular crystal observed in Figure 1i and Figure 3i is mullite whisker.
2MoSi_2_ + 7O_2_ = 2MoO_3_ + 4SiO_2_(R1)
3Al_2_O_3_ + 2SiO_2_ = 3Al_2_O_3_ ·2SiO_2_ (mullite)(R2)
SiC + 2O_2_ = SiO_2_ + CO_2_(R3)

As for the Si-based coating, cristobalite begins to crystallize, and the oxidation of MoSi_2_ occurs after the coating is calcined at 1000 °C. When the temperature increases to 1200 °C, the amount of cristobalite increases, and the oxidation of SiC occurs. When the Si-based coating is calcined at 1400 °C, mullite is generated, and the amount of cristobalite increases further. By comparison, mullite is produced at lower temperatures in the Al-based coating than in the Si-based coating, and more mullite is produced at the same temperatures because the nano alumina produced from the alumina sol has better activity than corundum. In the Al-based coating, most SiC and MoSi_2_ surfaces are in direct contact with the alumina gel and are coated homogeneously by a nanolayer of alumina. When SiC and MoSi_2_ are oxidized to produce SiO_2_, a direct interface between silica and alumina in the sol state decreases the diffusion resistance and consequently enhances the mullitization rate [8]. In addition, the cristobalite in Si-based coating shows lower production temperature and higher content than that in Al-based coating. This is mainly because the silica therein comes not only from the oxidation products of SiC and MoSi2 but also from the nano-silica in silica sol.

The FTIR spectra of the as-prepared Al-based coating and those calcined at 1000 °C to 1400 °C are shown in Figure 5a. The FTIR spectrum of the Al-based coating mainly reflects the properties of aluminum sol. The band around 1633 cm^−1^ is attributed to H-O-H bonds, and the vibrational band at 1080 cm^−1^ is indicating to the Al-O-H group’s existence in aluminum hydroxides (AlO(OH)) [23]. After calcined at 1000 °C, the H-O-H band weakens, and the Al-O-H band vanishes, suggesting the removal of bound water. Moreover, the stretching vibration of Si-O-Si of amorphous silica can be observed at 1111 cm^−1^, and the corresponding bending mode of Si-O-Si vibration is at 470 cm^−1^, suggesting the formation of SiO_2_ from the oxidation of MoSi_2_. After the calcination at 1200 °C, peaks at 595 cm^−1^ and 646 cm^−1^ appear due to the stretching mode of Al-O vibration in octahedral coordination [24]. The peak at 846 cm^−1^ also becomes obvious, indicating the formation of Si-O-Al linkages from the crystallized mullite phases [25]. On further heating to 1400 °C, the peak at 846 cm^−1^ becomes wider, indicating more crystallization of mullite with the formation of more Si-O-Al linkages in the mullite structure. Therefore, the evolution of the bonding phase in Al-based coating is as follows. In the initial stage of the sol-gel reactions, small three-dimensional oligomeric species are formed with Al-OH groups on their outer surface. Then, the linkage of the Al-OH groups occurs via polycondensation and eventually results in oxypolymers with the Al-O-Al network after drying, and the number of Al-O-Al bonds increases with time and temperature [26]. The oxidation of MoSi_2_ and SiC continuously produces highly reactive silicon oxide, which enters the three-dimensional network to form Si-O-Al. Finally, Al_2_O_3_ forms upon dehydration from the Al-O-Al, and mullite is generated from the Si-O-Al network [27].

The FTIR spectra of the as-prepared Si-based coating and those calcined at 1000 °C to 1400 °C are shown in Figure 5b. As for the spectrum of as-prepared Si-based coating, the very intense band appearing at 1117 cm^−1^ and the shoulder at around 1200 cm^−1^ are assigned to the transversal optical and longitudinal optical modes of the Si-O-Si asymmetric stretching vibrations, respectively. The peak near 471 cm^−1^ and a low-frequency peak near 576 cm^−1^ are assigned to Si-O-Si out-of-plane bending and Si-O-Si stretching modes, respectively [28,29]. The silica sol consists of hydrated silicon particles (Si-OH) and gels around the other solid particles during drying and provides the initial green strength when used as a binder. During initial drying, the hydroxyl group condenses to form a three-dimensional network of siloxane bonds (Si-O-Si), and the water molecule is removed from the structure in this process [13]. The peaks at 746 cm^−1^ indicating the formation of Si-O-Al linkages can be observed due to the binding of silica sol with nano alumina particles. After heating at 1000 °C, the peaks of Si-O-Si broaden and weaken, suggesting the transformation of silica gel networks to amorphous silicon oxide. When the calcination temperature is higher than 1200 °C, the bands at 620 cm^−1^ and 795 cm^−1^ appear, which are attributed to Si-O-Si symmetrical stretching vibrations of the individual SiO_4_ tetrahedra of cristobalite and the symmetric vibration of Si-O [24].

In order to determine the temperature of the chemical reactions and phase transitions in the Al-based coating and the Si-based coating, thermogravimetric analysis (TG) and differential scanning calorimetry (DSC) from room temperature to 1200 °C were conducted, and the results are shown in Figure 6a,b. As for the Al-based coating (Figure 6a), several small endothermal DSC peaks in the range of 50–300 °C with about 25.5% mass loss are observed, which could be ascribed to the removal of occluded water and hydrogen chloride. The alumina sol is prepared via the reaction of aluminum and hydrochloric acid, and the aluminum-chloride ratio is 1.3~1.4. Therefore, HCl is discharged from the coating during the heating process. As the temperature increases from 800 °C to 1200 °C, there is a broad endothermal peak and little change in mass, which is related to the reaction of SiO_2_ and Al_2_O_3_ to form mullite [30,31]. Compared with the Al-based coating, the Si-based coating shows significantly less weight loss (Figure 6b). The small endothermal peak at around 70 °C accompanied by a mass loss of 4.1% may be related to the discharge of free water. The broad endothermal peak at the range of 900–1200 °C also arises from the reaction of SiO_2_ and Al_2_O_3_ to form mullite. By contrast, the initial reaction temperature in Al-based coating (850 °C) is 75 °C lower than that in the Si-based coating, and this is consistent with the XRD results in Figure 4.

### 3.3. Emissivity of the Coating

The emissivity of the coatings determines its ability to radiate heat, and the total radiation exitance is proportional to the emissivity. The spectral reflectivity of the Al-based coating and Si-based coating are shown in Figure 7a,c at the wavelength range of 200 nm–25 μm, and both the fully dried coatings and the coatings calcined at 1200 °C for 1 h are tested. The accordingly calculated spectral emissivity of the coatings is presented in Figure 7b,d. The total emissivity of the coatings calculated according to Equation (1) are listed in Table 2.

The total emissivity of the fully dried Al-based coating and Si-based coating are 0.922 and 0.892, respectively, at 200–2500 nm, and are 0.897 and 0.906, respectively, at 2.5–25 μm. The spectral emissivity of Al-based coating is higher than that of Si-based coating at a wavelength lower than 10 μm. Since the emissivity in the low wavelength range is mainly affected by the emissive agents [32], and the emissive agents in the two coatings are the same, this difference may be related to the surface roughness. The SEM images in Figure 1 and Figure 3 suggest that the Al-based coating has a rougher surface than the Si-based coating, which leads to absorptivity increase and reflectivity decrease by the small gradual slopes in the grooves [33,34]. The significant absorption band centered at around 9 μm related to the asymmetrical Si-O stretching vibrations is avoided in the Al-based coating, which makes the emissivity in this band increase significantly. The higher emissivity in the low wavelength range of Al-based coating will increase the total radiation exitance at high temperatures because as the temperature increases, the peak wavelength of radiation will gradually move toward the shorter wavelength according to Wien’s displacement law [4]. By contrast, the spectral emissivity of Al-based coating is slightly lower than that of the Si-based coating at a wavelength higher than 10 μm because of the more characteristic absorption peaks associated with Al-O bonds [35].

When the Al-based coating and Si-based coating are calcined at 1200 °C for 1 h, the spectral emissivity decreases in the low wavelength range (<6 μm) and increases in the high wavelength range (>10 μm). The decreased emissivity in the low wavelength range is mainly due to the oxidation of the emissive agents [36,37], and the oxidation of MoSi_2_ and SiC at 1200 °C has been verified by the XRD analysis. By contrast, the emissivity reduction after calcination at 1200 °C of the Al-based coating (−10.2%) is smaller than that of the Si-based coating (−18.6%), and the emissivity of the whole band is higher than 0.82, which make the Al-based coating more suitable for use in a high-temperature oxidation environment. The increased emissivity at a high wavelength range after calcination is possibly related to the surface structure changes and the formation of mullite.

### 3.4. Thermal Shielding and Insulation Performance

The thermal shielding and insulation performance of the Al-based coating is evaluated by the cold side temperature when the material is directly exposed to the flame of a butane blow torch, and the photo of the experiment is presented in Figure 8a. The maximum temperature of the flame can reach 1300 °C, which remains almost constant under stable butane intake and sufficient air conditions. The cold side temperature curves over time of the Al-based coated substrate and bare substrate when the substrate thickness is 15 mm, which is shown in Figure 8b. The cold side temperature rises slowly from room temperature, eventually reaching a stable temperature at around 400 s. The stable cold side temperature of the bare substrate and the Al-based coated substrate are 365 °C and 335 °C, respectively. The cold side temperature of the bare substrate is significantly lower than the flame temperature, which is mainly due to the thermal insulation effect of the low thermal conductivity substrate (0.0995 W/(m·K)). The cold side temperature of the Al-based coated substrate is 30 °C lower than that of the bare substrate, which is related to the radiation heat dissipation of the high emissivity coating. This difference is further amplified when the substrate thickness is reduced to 8 mm, and the stable cold side temperature of the bare substrate and the Al-based coated substrate are 680 °C and 465 °C, respectively (Figure 8c). According to the Fourier heat equation for heat conduction in one dimension: $\frac{dq}{dt}=\lambda \frac{dT}{dx}S$, where $\frac{dq}{dt}$ is the heat conduction per unit time, λ is the thermal conductivity coefficient, *S* is the sectional area, and $\frac{dT}{dx}$ is the temperature gradient. Under the condition of constant thermal conductivity, the hot and cold side temperature difference is proportional to the thickness of the insulation material [38,39], so the cold side temperature of the bare substrate increases significantly when the thickness is reduced from 15 mm to 8 mm. The radiant heat dissipation of the coating is proportional to the emissivity and the fourth power of the temperature [40], so its effect is more remarkable under the condition of a thin insulation substrate. It should be noted that the appearance of the sample has not changed significantly after the surface flame test, suggesting that the Al-based coating possesses good temperature resistance and stability.

## 4. Conclusions

In this study, aluminum sol was used as a binder and MoSi_2_ and SiC as emittance agents to prepare a high emissivity MoSi_2_-SiC-Al_2_O_3_ coating on the surface of mullite insulation tiles. The evolution of structure and composition, as well as the spectral emissivity of the coating, was compared with a silica-sol-bonded coating with the same refractory fillers and emissive agents, and the main results are as follows.

(1)The as-prepared Si-based coating shows a denser structure, while the Al-based coating is porous, and the surface is rougher. After high-temperature heat treatment, the coating surface forms a network of micro-cracks, but the cracks are finer, and the network is more uniform in the Al-based coating. The more stable structure of Al-based coating with the increase in temperature increases the temperature resistance of the coating and increases the safety of service greatly.(2)When calcined at 1400 °C, an acicular mullite whisker is formed on the coating surface, but when the generation temperature is lowered by 75 °C, the whisker yield is significantly higher in the Al-based coating. The formation of mullite whisker is expected to improve the thermal shock resistance of the coating and its matching with the substrate.(3)The total emissivity of the as-prepared Al-based coating is 0.922 and 0.897 at the wavelength range of 200–2500 nm and 2.5–25 μm, respectively. The superior emissivity at low wavelength (<10 μm) is related to the higher surface roughness and reduced feature absorption. The spectral emissivity of the coatings calcined at 1200 °C decreases in the low wavelength range (<6 μm) due to oxidation of the emissive agents, and the emissivity reduction in the Al-based coating (−10.2%) is smaller than that of the Si-based coating (−18.6%). The high emissivity and low emissivity degradation at high temperatures make the Al-based coating an excellent radiative heat protection coating.(4)A flame heating experiment shows that the cold surface temperature of the Al-based coating coated substrate is 30 °C and 215 °C lower than the bare substrate when the substrate thickness is 15 mm and 8 mm, respectively. Overall, the Al-based coating produces good temperature resistance, high emissivity, and excellent thermal insulation performance.

## Figures and Tables

**Figure 1 materials-17-00220-f001:**
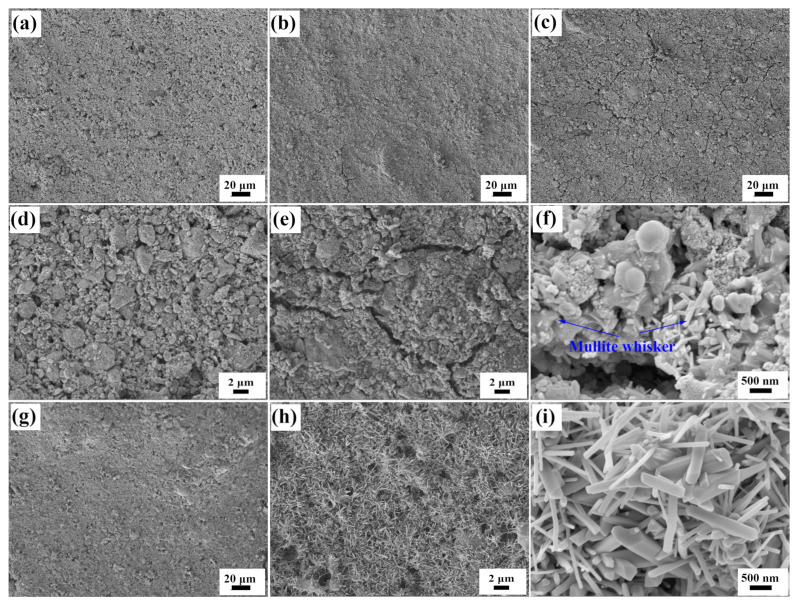
SEM images of the surface of Al-based coating (**a**,**d**) and those calcined at high temperatures (1000 °C (**b**,**e**), 1200 °C (**c**,**f**) and 1400 °C (**g**–**i**)).

**Figure 2 materials-17-00220-f002:**
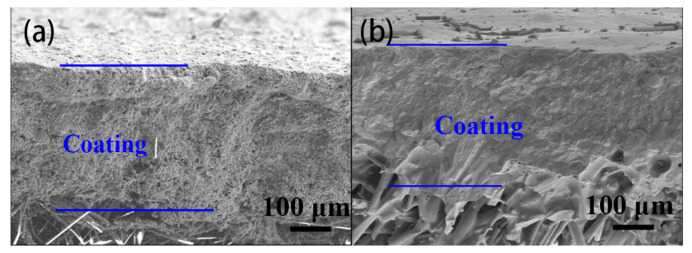
The cross-sectional SEM images of the Al-based coating (**a**) and the Si-based coating (**b**).

**Figure 3 materials-17-00220-f003:**
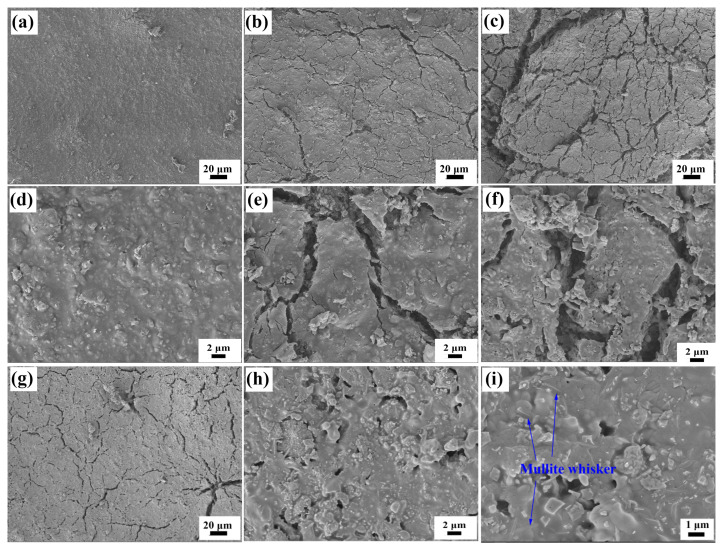
SEM images of the surface of Si-based coating (**a**,**d**) and those calcined at high temperatures (1000 °C (**b**,**e**), 1200 °C (**c**,**f**) and 1400 °C (**g**–**i**)).

**Figure 4 materials-17-00220-f004:**
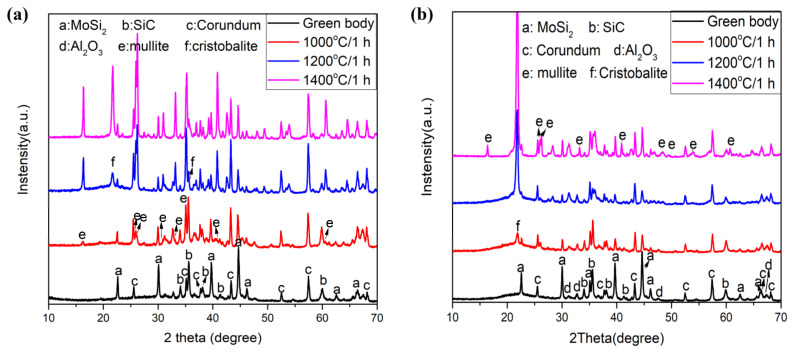
The XRD patterns of the Al-based coating (**a**) and Si-based coating (**b**) after calcination at 1000–1400 °C.

**Figure 5 materials-17-00220-f005:**
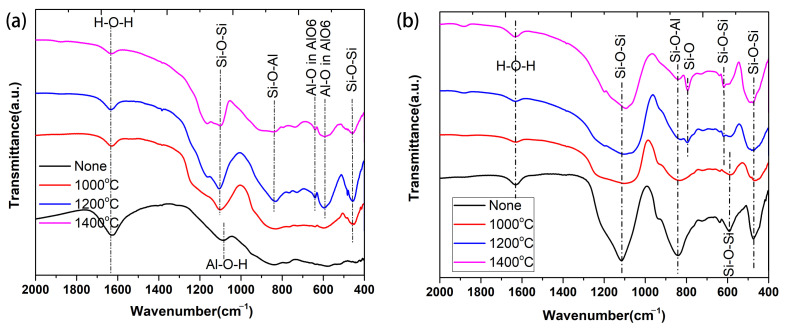
The FTIR spectra of the Al-based coating (**a**) and Si-based coating (**b**) after calcination at 1000–1400 °C.

**Figure 6 materials-17-00220-f006:**
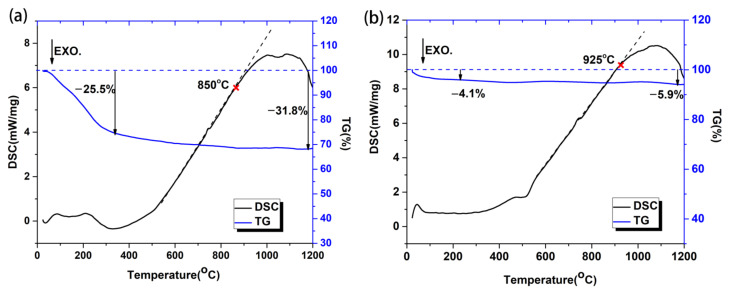
The TG/DSC curves of the Al-based coating (**a**) and the Si-based coating (**b**) from room temperature to 1200 °C.

**Figure 7 materials-17-00220-f007:**
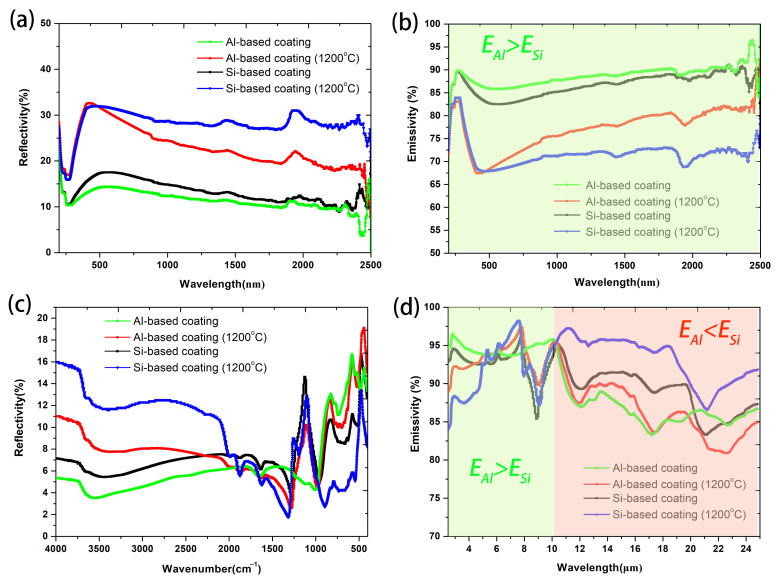
Spectral reflectivity (**a**,**c**) and emissivity (**b**,**d**) of the coatings at the wavelength range of 200–2500 nm (**a**,**b**) and 2.5~25 μm (**c**,**d**).

**Figure 8 materials-17-00220-f008:**
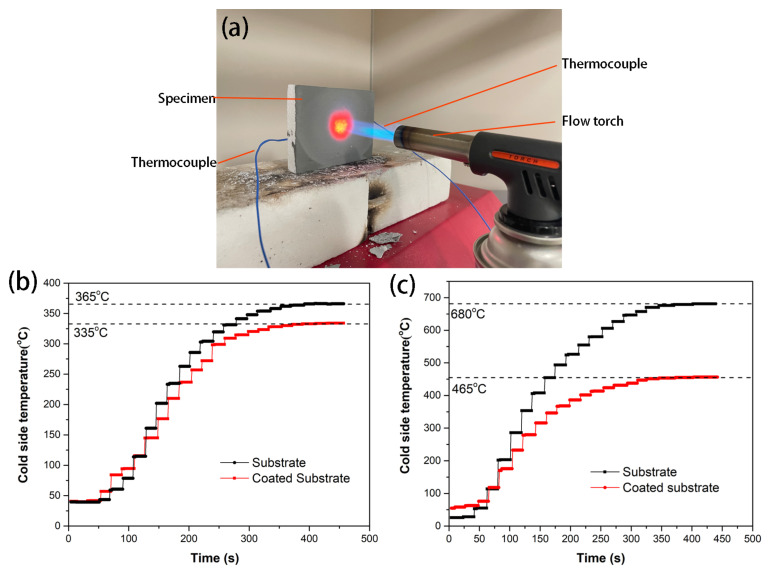
Photo of the flame heating experiment (**a**) and the cold side temperature curves over time of the Al-based coating coated substrate and bare substrate: the thickness of the substrate is 15 mm (**b**) and 8 mm (**c**).

**Table 1 materials-17-00220-t001:** Properties of silica sol and alumina sol in this work.

Items	pH	Solid Load (%)	Viscosity (mPas)	Density (g/cm^3^)	Particle Size (nm)
Silica sol	5.2	30	<7	1.21	10–20
Alumina sol	4.3	20	<7	1.31	20–30

**Table 2 materials-17-00220-t002:** Total emissivity of the coatings.

Emissivity	200–2500 nm		2.5–25 μm	
	Fully Dried	1200 °C/1 h	Fully Dried	1200 °C/1 h
Al-based coating	0.922	0.828	0.897	0.897
Si-based coating	0.892	0.726	0.906	0.939

## Data Availability

Data are contained within the article.

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
