# Peer review of "High Emissivity MoSi2-SiC-Al2O3 Coating on Rigid Insulation Tiles with Enhanced Thermal Protection Performance"

_materials, 2023, doi:10.3390/ma17010220_

Round 1

Reviewer 1 Report

Comments and Suggestions for Authors

The study is well conducted and the manuscript is well written, some issues marked in the attached file should be corrected before acceptance of the manuscript

Comments on the Quality of English Language

English is very good, some errors are indicated in the attached file

Author Response

Thanks a lot for your comments and the response please see the attachment. 

Reviewer 2 Report

Comments and Suggestions for Authors

The paper "High emissivity MoSi2-SiC-Al2O3 coating on rigid insulation tiles with enhanced thermal protection performance" is devoted to the study of high emissivity MoSi2, SiC and Al2O3-based coatings used on rigid insulation boards. The authors study the evolution of the structure and composition of the coating at high temperatures, as well as its spectral emission and thermal insulation properties. By comparing the coatings using silicate salt, the authors find that the aluminum salt-based coating has a more homogeneous structure, greater surface roughness, and is able to generate mullite whiskers at lower temperatures. The emissivity of the coating is 0.922 and 0.897 in the wavelength ranges of 200–2500 nm and 2.5–25 μm, respectively. The study also indicates a relatively small decrease in emissions upon oxidation of emissive agents at high temperatures. The authors conclude by emphasizing the excellent thermal insulation properties of the coating.

The paper can be accepted after the following comments:

1.    The paper is well structured and clearly describes the purpose of the study, the methodology, and the results obtained. However, it is recommended that the authors pay more attention to the introduction to justify the relevance of the study and to attract the reader's interest.

2.    In the Materials and Methods section, a more detailed description of the parameters and conditions of the experiment should be provided. This will allow other researchers to repeat the experiment and obtain comparable results.

3.    Based on the data presented in Figure 3, it is recommended that the Rietveld method be applied to determine the phase content of the MoSi2-SiC-Al2O3 coating. This will provide more accurate and quantitative information on the structure and composition of the coating, which may be useful for a better understanding of its properties and possible applications.

4.    Based on the data presented and the results of SEM and XRD analysis, it will be possible to evaluate the effect of temperature on the morphology and structure of the coating and to determine whether the use of a gas torch is sufficient for the required temperatures or whether the use of a hydrogen torch may be more effective.

5. The study lacks a comparative table with methods of producing ceramic materials such as HIP (hot isostatic pressing), SPS (pulsed current rapid sintering), MW (microwave sintering), vacuum sintering, and others (10.1134/S0036023618110177, 10.1134/S0036023621050168, 10.1134/S1087659618060159).

Author Response

(The authors gave the same response as above.)

Reviewer 3 Report

Comments and Suggestions for Authors

Introduction: Please highlight the goal of the paper.

Materials and Methods: What was the thickness of the coating? Did you use the same thickness for both sample dimensions (100 mm×130 mm × 8 mm and 100 mm×130 mm × 15 mm)? In 3.1. Coating structure chapter you pointed thickness of coating to be around 300 μm, you could add the dimensions of the samples in privius chapter. Also, do you have some measurements of the coating thickness, and average values, compared to Si coatings and their thickness uniformity.

Also, did you have some data about coating/mulite bonding. Are there some differences in adhesive bonding using Si or Al based coatings? Do you have some photos of mulite/coatings inteface? How the calcination temperature influence on interface?

At the figure 1. only surface was monitored. Do you have figs. of the cross section?

Fig.1.  analysis : Are the placement of crack formation closer to surface or interface. Do you have figures of cross-section of the sample? Could the placement of the crack could be related to the calcinations temperature?

Fig.1.a is related to the interface, but maybe it is better to give the separate figure.

Fig.4. Please add the bar/ number to Temperature axe.

Conclusion: the main conclusions could be given as bullets or numbering.

Figure 4. Please, add the bar /number to the Temperature axe.

Comments on the Quality of English Language

Introduction: Please highlight the goal of the paper.

Materials and Methods: What was the thickness of the coating? Did you use the same thickness for both sample dimensions (100 mm×130 mm × 8 mm and 100 mm×130 mm × 15 mm)? In 3.1. Coating structure chapter you pointed thickness of coating to be around 300 μm, you could add the dimensions of the samples in privius chapter. Also, do you have some measurements of the coating thickness, and average values, compared to Si coatings and their thickness uniformity.

Also, did you have some data about coating/mulite bonding. Are there some differences in adhesive bonding using Si or Al based coatings? Do you have some photos of mulite/coatings inteface? How the calcination temperature influence on interface?

At the figure 1. only surface was monitored. Do you have figs. of the cross section?

Fig.1.  analysis : Are the placement of crack formation closer to surface or interface. Do you have figures of cross-section of the sample? Could the placement of the crack could be related to the calcinations temperature?

Fig.1.a is related to the interface, but maybe it is better to give the separate figure.

Fig.4. Please add the bar/ number to Temperature axe.

Conclusion: the main conclusions could be given as bullets or numbering.

Figure 4. Please, add the bar /number to the Temperature axe.

Author Response

(The authors gave the same response as above.)

Reviewer 4 Report

Comments and Suggestions for Authors

The manuscript is well written, and the research design is appropriate. I have a few minor comments that will further enhance the paper for the readers-

1.      To strengthen the paper's contribution, consider explicitly connecting your findings to those of similar studies in the field. Providing comparative insights could highlight the uniqueness of your results and their potential implications.

2.      It would be valuable to correlate the observed structural changes in the coatings with enhanced functional properties.

3. If possible, please quantify the observed differences in crystalline phases between the Al-based and Si-based coatings by comparing the relative abundance of each phase. This approach has been successfully employed in recent.

4 Please do not overlap the figure graphs with the legend.

Author Response

(The authors gave the same response as above.)

Round 2

Reviewer 2 Report

Comments and Suggestions for Authors

Thank the authors for making additions and accompanying the comments made with appropriate and compelling responses. The paper may be accepted in its current form.

Reviewer 3 Report

Comments and Suggestions for Authors

in the present state, after accepted comments incorporated in manuscript, the paper could be accepted for publication.